# Diversity of Culturable Yeasts Associated with the Technification Level in the Process of Mezcal Production in the State of Durango

Sandra Consuelo Martínez-Estrada [1], José Alberto Narváez-Zapata [2], Raúl Rodríguez-Herrera [3],
Julio Grijalva-Ávila [1], José Natividad Gurrola-Reyes [1], Claudia Patricia Larralde-Corona [2,*]
and Isaías Chairez-Hernández [1,*]

[1] CIIDIR-Durango-Instituto Politécnico Nacional, Calle Sigma #119, Fracc. 20 de Noviembre II,
Durango 34220, Durango, Mexico; con_sandr@hotmail.com (S.C.M.-E.); jcgrijalva69@gmail.com (J.G.-Á.);
natigre1@hotmail.com (J.N.G.-R.)

[2] Laboratorio de Biotecnología Industrial, Centro de Biotecnología Genómica-Instituto Politécnico Nacional,
Blvd. del Maestro s/n Esq. Elías Piña, Col. Narciso Mendoza, Reynosa 88700, Tamaulipas, Mexico;
jnarvaez@ipn.mx

[3] Departamento de Investigación en Alimentos, Facultad de Ciencias Químicas, Universidad Autónoma de
Coahuila, Blvd. V. Carranza y José Cárdenas s/n Col. República Oriente, Saltillo 25260, Coahuila, Mexico;
raul.rodriguez@uadec.edu.mx

\* Correspondence: plarralde@ipn.mx (C.P.L.-C.); ichairez@ipn.mx (I.C.-H.);
Tel.: +52-55-57296000-x-87726 (C.P.L.-C.); +52-618-8144540 (I.C.-H.)

**Abstract:** Durango State has the denomination of origin for the production of mezcal, which is made from *Agave durangensis*, mainly in an artisanal way; therefore, differences in the fermentation process affect the quality of the final product. The main objective of the present study was to evaluate the diversity of culturable yeasts involved in the artisanal and semi-technified process of mezcal production in the State of Durango. Three distilleries with different production processes were monitored at different fermentation stages (beginning, mid-fermentation, and end of fermentation) in the spring and summer seasons. A greater diversity was found in the distillery of Nombre de Dios in both the spring and summer production seasons (H′ = 1.464 and 1.332, respectively), since it maintains an artisanal production process. In contrast, the distillery of Durango, where a *Saccharomyces cerevisiae* commercial inoculum is used to start fermentation, presented low diversity indexes (H′ = 0.7903 and 0.6442) and only *S. cerevisiae*, *Kluyveromyces marxianus*, and, sporadically, *Pichia manshurica* were found. Results suggest that the yeast microbiota involved in mezcal fermentation during the different seasons is affected by the type of inoculum; changes include the presence of some species that were only identified during a specific season in alcoholic fermentation, such as *Torulaspora delbrueckii* and *Pichia kluyveri*.

**Keywords:** culturable yeast diversity; fermentation; mezcal; *Agave durangensis*; *Saccharomyces*; non-*Saccharomyces*

## 1. Introduction

Mexico is recognized for its production of high-quality alcoholic beverages such as tequila and mezcal, which are obtained by distilling the fermented juice of different species of *Agave* [1–3]. In the State of Durango, mezcal is made from *Agave durangensis*, which is the dominant agave species in the region. In this state, mezcal production is mainly artisanal and is protected by a denomination of origin [4]. Given that the agave that is used must be at least eight years old, the beverage production process begins by obtaining the raw material. First, leaves are removed from the agave plant [5], which results in a round stem (piña) of several kilograms, which is then cooked; during the cooking process, material is softened, since the biopolymer agavin and other fructooligosaccharides are hydrolyzed into simple sugars, mainly fructose [6]. Once the piña is cooked, it is crushed

and pressed to obtain a very dark (Maillard products) agave must, which is then diluted and placed in the fermentation vats, and then allowed to ferment without (spontaneous fermentation) or with a commercial *Saccharomyces cerevisiae* inoculum (frequently, a bread strain), depending on the producer (Figure 1). Agave must fermentation is carried out to produce ethanol and other compounds that define the main characteristics of mezcal, such as organoleptic compounds [1]. Most studies only describe the process of mezcal production; however, Kirchmayr et al. conducted a study to elucidate the impact of changes in the process and thereby improve the fermentation efficiency; they found that changes in the production process impact microbial diversity, in addition to the fact that the use of inoculums increases the efficiency of alcoholic fermentation [7]. However, changes in volatile compounds must also be considered; volatile compounds may vary depending on the raw material, production process, geographic region, climatic conditions, and, therefore, the microflora involved in this process [8,9].

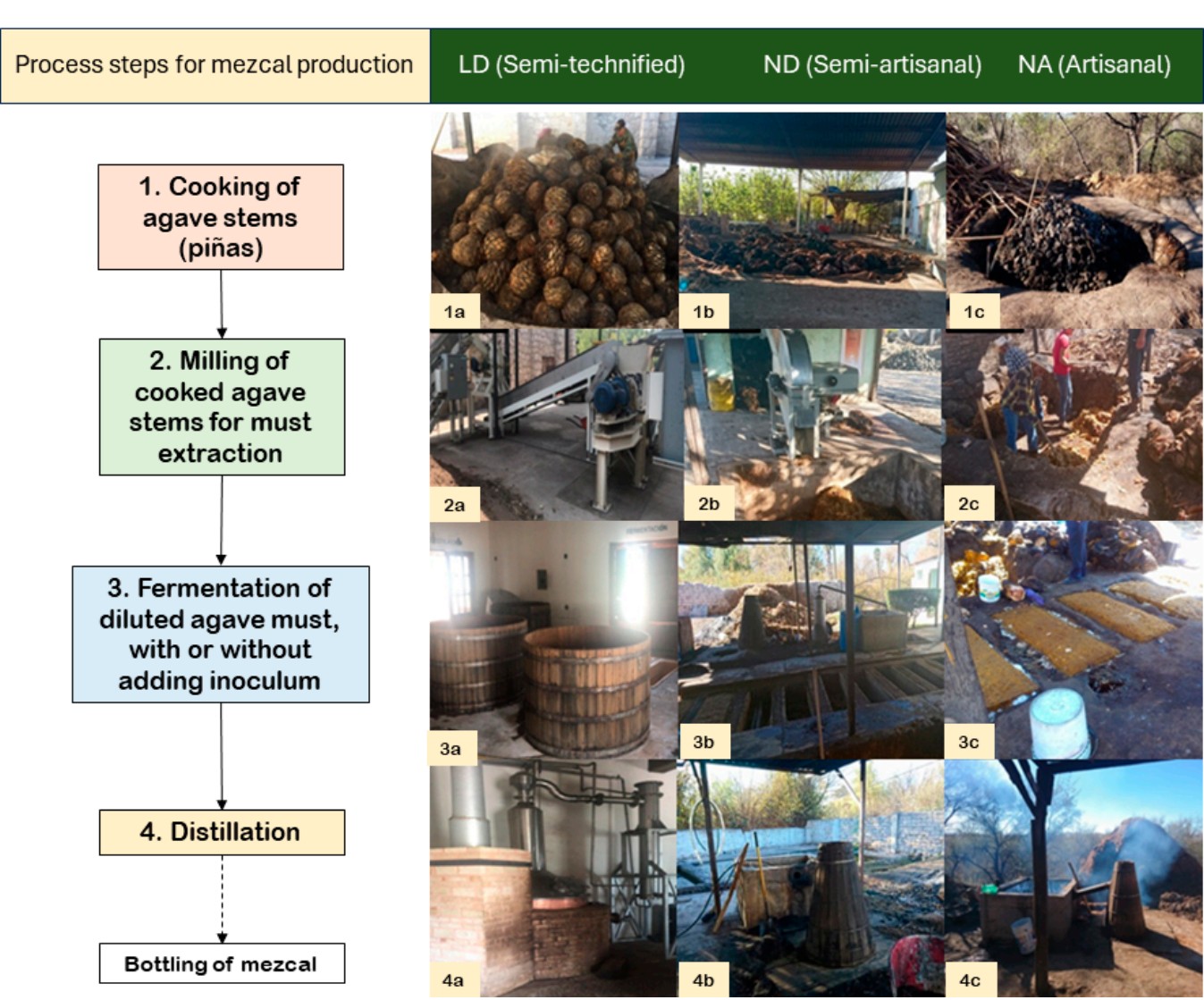

**Figure 1.** Process of elaboration of mezcal in the three different distilleries, where all (**1a**–**4a**) are from the Municipality of Durango distillery (LD), which is semi-technified, all (**1b**–**4b**) are from La Constancia distillery (ND), and all (**1c**–**4c**) are from Nombre de Dios distillery (NA). Type of cooking: (**1a**) conical oven; (**1b**) and (**1c**) earth oven. Technique for must extraction: (**2a**) milling train, (**2b**) mechanical mill, (**2c**) manual milling with an ax. Fermentation vats: (**3a**) wooden barrel; (**3b**) and (**3c**) earthed wooden containers. Distilling equipment: (**4a**) stainless steel stills; (**4b**) and (**4c**) copper alambique and wood cooler.

Studies have been conducted on the identification of the microorganisms involved in the production of mezcal. Escalante-Minakata et al. identified three species of yeasts in the fermentation of *Agave salmiana* from San Luis Potosí State, as follows: *Clavispora lusitaniae*, *Pichia fermentans*, and *Kluyveromyces marxianus* [10]. Verdugo-Valdez et al. found a greater yeast diversity in the fermentation of *A. salmiana* of the same state and reported the presence of *S. cerevisiae*, *K. marxianus*, *Pichia kluyveri*, *Zygosaccharomyces bailii*, *Clavispora lusitaniae*, *Torulaspora delbrueckii*, *Candida ethanolica*, and *Saccharomyces exiguous* [11]. In Durango, Páez-Lerma et al. found *S. cerevisiae*, *K. marxianus*, *Candida diversa*, *T. delbrueckii*, *P. fermentans*, and *Hanseniaspora uvarum* at the beginning of the fermentation, but only *S. cerevisiae* and *T. delbrueckii* at the end; additionally, they demonstrated that *S. cerevisiae* strains of the State of Durango are phylogenetically independent of strains isolated from regions of Latin America and Europe [4]. In the State of Oaxaca, Kirchmayr et al. analyzed the impact of the process modifications in two distilleries in two consecutive years, identifying *S. cerevisiae*, *K. marxianus*, *Zygosaccharomyces rouxii*, *Zygosaccharomyces bisporus*, *T. delbrueckii*, and *Pichia membranaefaciens* as the most frequent yeast species in the first distillery, while, in the second, *T. delbrueckii* and *Z. bisporus* were found [7]. Additionally, they observed that the number of *S. cerevisiae* isolates increased by 28% due to its abundance in the inoculum used, decreasing the Shannon–Wiener index (2.223 to 1.509).

Villarreal Morales et al. identified metagenomic populations in the mead of *Agave salmiana* and *A. atrovirens* during the four seasons of the year and found greater diversity in the summer season in samples from both agave species (Shannon–Wiener index: 2.43 in the summer and 1.93 in the rest of the seasons), identifying *K. marxianus*, *S. cerevisiae*, and *Kazachtania zonata* [12]. Enríquez-Salazar et al. analyzed the cultivable microbial diversity in *A. salmiana* and *A. atrovirens* mead, finding greater biodiversity during the winter and summer seasons (Shannon–Wiener index 2.1 in the winter and 2.01 in the summer) [13]. Due to the variations in fermentation conditions, it is common to find mezcal with different qualities, so it is essential to start the fermentation using identical inoculums to provide homogeneous quality between the lots. Since the process of producing mezcal in the State of Durango is usually carried out by spontaneous fermentation, microorganism presence is not controlled, and a large complex of natural microorganisms is usually involved [14]. The objective of this work was to identify and evaluate the diversity of cultivable yeasts during the different fermentation stages (beginning, middle, and end) of *A. durangensis* must from three producing distilleries with different levels of technification in the State of Durango, in two different seasons, to assess the influence of such parameters in the population dynamics of yeasts during the fermentation processes.

## 2. Materials and Methods

### 2.1. Selected Distilleries by Technification Level

The production process of three distilleries in the State of Durango was characterized. The selected locations were as follows: one in the Municipality of Durango (LD) and two in the Municipality of Nombre de Dios, but in different towns, including one located in La Constancia (ND) town and one located in Nombre de Dios (NA) town. Selected distilleries use the same process steps, but different techniques and materials (technification level) for mezcal elaboration. According to the specific process used, distilleries ND and NA are classified in the category of artisanal mezcal, although ND uses a mechanical mill (hence, referred to here as a semi-artisanal production), while LD is classified as semi-technified mezcal [15]. Both ND and NA maintain a spontaneous fermentation, unlike LD, which uses commercial inoculum of *S. cerevisiae* to initiate it. In Durango State, mezcal is made from *A. durangensis*, which grows in the wild; ND and NA distilleries collect the agave from communal lands, while LD collects it from a private state property in the municipality of El Mezquital. The differences between ND and NA processes are depicted in Figure 1.

NA uses a conical earth oven to cook the agave; then, the cooked agave is crushed manually with an ax, the must fermentation is placed in wooden vats, and the distillation takes place in a copper vessel covered with wooden still; NA production is oriented to bulk

marketing in the same production area. The ND distillery uses a conical earth oven to cook the agave, which is then mashed with a mechanical mill, and, as in NA, fermentation takes place in wooden vats and distillation in a copper vessel with a wooden still; production is commercial. In both factories, firewood is used as fuel, and the processes are carried out according to a traditional recipe passed through the owners' generations. In the third distillery (LD), agave steams are cooked in a conical earth oven, mashed with a mill and a press, and the fermentation is carried out in wooden vats with a capacity of 1700 L; for the distillation, copper and stainless-steel stills are used with LP gas as fuel. In LD, the head of the production is a biochemical and biotechnology engineer with a specialization in fermentation. LD produces 800 to 1500 L per month, and part of the production is exported.

### 2.2. Sample Collection

A fermentation must sample of 100 mL was taken directly from the fermentation vats, collecting liquid from the bottom, middle, and upper positions to have an average sample from the whole fermentation. Three different monitoring times were used (beginning, middle, and end of the fermentation process) to consider the different fermentation times from every distillery: NA and ND established their fermentation duration according to foam formation, while LD tracked sugar consumption (Table 1). Sampling took place during the spring and summer seasons. The NA location, Nombre de Dios, presented an average rainfall of 19.75 mm, a maximum temperature of 27 °C, and a minimum temperature of 11.5 °C, with June as the warmest month with a maximum of 30 and a minimum of 15 °C. In the ND location, La Constancia town, average rainfall was 18.25 mm, maximum and minimum temperatures were 26.25 and 11.75 °C, and June was the warmest month with a 29 °C and 15 °C maximum and minimum temperatures. In the LD location, average rainfall was 9.75 mm, the maximum temperature was 26.25 °C, while the minimum temperature reached 10.25 °C, and during the warmest month, June, the maximum and minimum temperatures were 29 and 14 °C, respectively.

**Table 1.** Level of technology and geographical coordinates of the location of the mezcal companies.

| Distillery | Code | Technification Level | Altitude (masl) | Latitude | Longitude |
|---|---|---|---|---|---|
| Durango | LD | Semi-technified | 1860 | 24° 01.388 | 104° 56.833 |
| La Constancia | ND | Semi-artisanal | 1760 | 23° 91.611 | 104° 26.583 |
| Nombre de Dios | NA | Artisanal | 1740 | 23° 84.361 | 104° 24.083 |

### 2.3. Yeast Count and Isolation

Must samples were diluted according to Mexican regulation [16]. Bacteria and yeasts were counted in Luria–Bertani (LB), nutrient, and potato dextrose agar (PDA) media by plating in triplicate. For bacteria growing, plates were incubated for 48 h at 35 ± 2 °C; for yeast growing, plates were incubated at 25 ± 1 °C for 5 days. Bacteria and yeast colony-forming units (CFUs) were counted by following the Mexican Official Norm specifications [16,17]. Yeast isolates were purified by a subculture for later analysis.

### 2.4. DNA Extraction and PCR Amplification of the D1/D2 LSU Regions

Genomic DNA extraction was performed with a Wizard® Genomic DNA Purification Kit (PROMEGA, Madison, WI). D1/D2 LSU regions (26S region) were amplified through polymerase chain reaction (PCR) with NL1 (5′ GCATTCAATAAGCGGAGGAAAAG 3′) and NL4 (5′ GGTCCGTGTTTCAAGACGG 3′) primers [18]. For the amplification reaction, 2.5 mM of each primer, 15μM dNTPs, 1.5 mM MgCl2, 1X buffer, and 1U of Taq polymerase were suspended in 31.5 μL of distilled and deionized water. The conditions were: 1X (94 °C for 5 min), 35 cycles (94 °C for 30 s, 61 °C for 30 s, 72 °C for 1 min), 1X (72 °C for 7 min), and 1X (6 °C for 5 min). Amplified fragments were visualized in 1% agarose gel electrophoresis in 1X TAE buffer. The 26S rRNA gene-amplified fragments were sequenced through the

ABI PRISM® 310 Genetic Analyzer (Thermo Fisher Scientific, Waltham, MA, USA) assisted by the Bright Dye® Terminator Cycle Sequencing kit (McLab, San Francisco, CA, USA) using the forward primer for the 26S rRNA region.

### 2.5. Identification of Microorganisms and Phylogenetic Analysis

Sequences were edited with ChromasPro v2.1.8 (Technelysium Pty Ltd., South Brisbane, Australia); edited sequences were compared to the GenBank database using the BLAST program (http://blast.ncbi.nlm.nih.gov, accessed on 5 December 2023) to determine the nucleotide identity of the isolates and obtain reference sequences. Sequences were then aligned to GenBank using CLUSTALW v2.0 [19]. The phylogenetic tree was constructed by using the maximum likelihood algorithm. The ME tree was searched using the Close-Neighbor-Interchange (CNI) algorithm, using 1000 rapid bootstrap inferences in the MEGA11 software version 11 [20]. The trees were visualized and edited iTOL software version 6 (https://itol.embl.de/, accessed on 5 December 2023).

### 2.6. Diversity Analysis

Alfa diversity for yeasts populations was determined through ecologic parameters with PAST v2.17 software [21] by applying the diversity index (Shannon–Wiener index, H'); the dominance index (Simpson index, D) and species richness index (Margalef index, DMg) were used to observe low- and high-diversity zones in the studied species from the samples. Indices were estimated with 1000 bootstrap repetitions. A Principal Component Analysis (PCA) was conducted considering the isolates by genus, and as variables, the different fermentation times in each distillery (NA, ND, and LD) within each sampling season (spring or summer). Cos2 values were considered to evaluate the quality and contribution of the sample variables [22]. The package "Factoextra" in RStudio was used to conduct this analysis.

### 2.7. Analytical Techniques

Must samples were analyzed to determine the reducing sugars concentration, protein content, density, pH, refraction index, and dissolved solids. Reducing sugars were determined by the 3,5-dinitrosalicylic acid method [23] using a glucose standard curve; protein determination was performed by the Kjeldahl method; the must density was determined employing a densimeter; the pH determinations were made with a three-point calibrated potentiometer; the refractive index was determined by hand refractometer; and total dissolved solids were measured as ppm [24]. The experimental design used was a factorial array with three levels and three repetitions. Factors were the sampling season and production technification level; response variables were the must temperature, fermentation stage (beginning, middle and end), bacteria CFUs in two culture media, yeast CFUs, protein percentage, density, pH, reducing sugars content, refraction index, and the ppm of the dissolved solids. Data were entered into SAS 9.0 software (SAS Institute, Cary, NC, USA) to perform an ANOVA ($p \leq 0.05$); a Tukey test was performed to compare treatment means.

## 3. Results

### 3.1. Yeast Identification

We isolated 143 yeast colonies from PDA agar plates and identified the species level by sequencing the D1/D2 LSU region; eight yeast species from seven different genera were genetically identified. The yeast species identified were *S. cerevisiae*, *K. marxianus*, *Z. bailii*, *P. manshurica*, *T. delbrueckii*, *P. kluyveri*, *Hanseniaspora guilliermondii*, and *C. lusitaniae* (62 sequences reported on the GenBank, from MT322329.1 to MT322391.1). Phylogenetic analysis showed a clear separation among the phylogenetic taxa of the eight species found in *A. durangensis* fermentation, including between the *Pichia* species *P. manshurica* and *P. kluyveri*. Reference organisms with their GenBank numbers were included to clarify the taxa location (Figure 2).

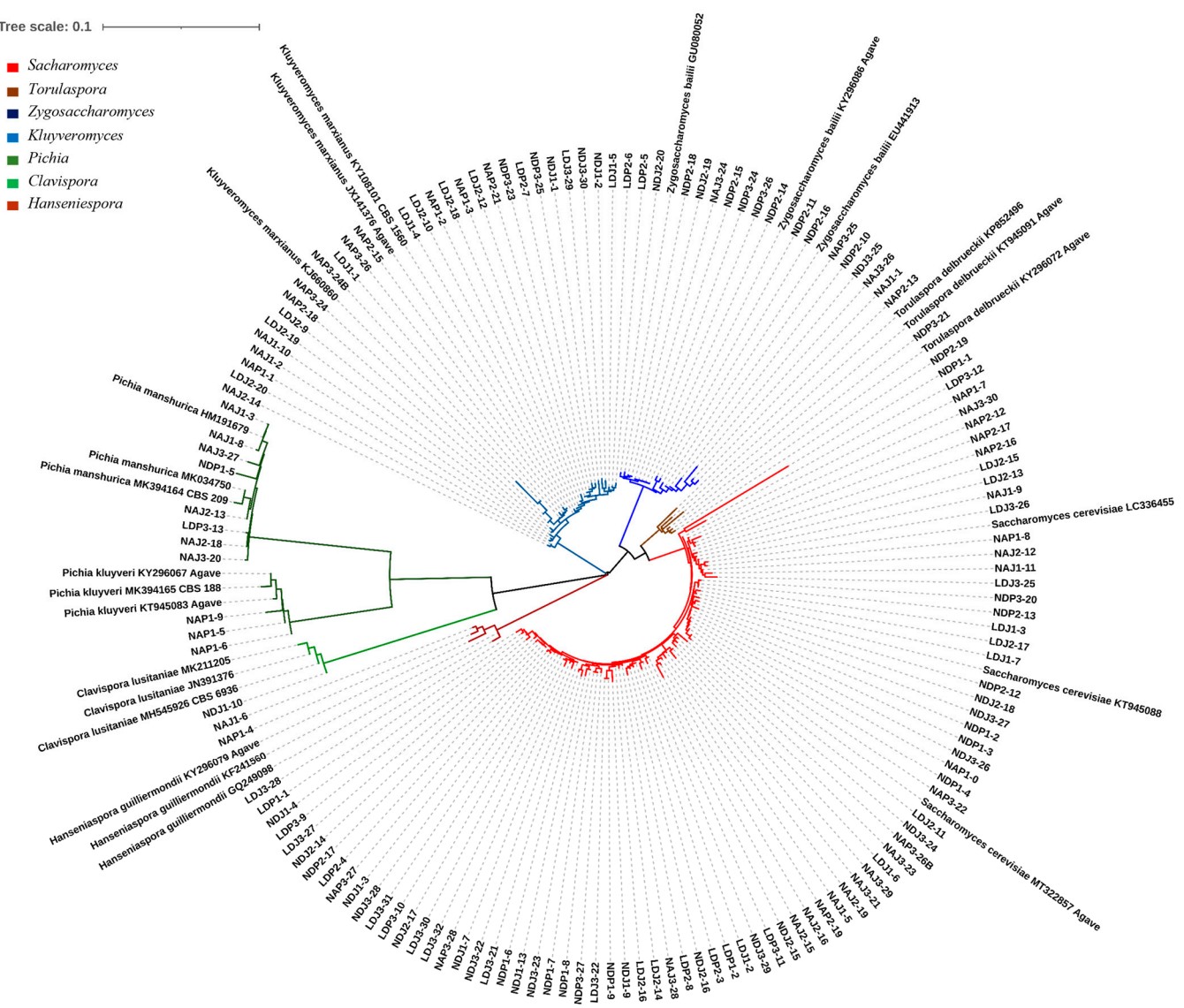

**Figure 2.** Phylogenetic relationship of the D1/D2 LSU region of strains isolated from mezcal fermentation and reference sequences. Colors indicate the different genera identified. Accessions numbers are indicated at the end of the reference strains.

The most frequently isolated yeast species was *S. cerevisiae*, found in samples from the three distilleries and at the three fermentation times (Figure 3). Yeast *S. cerevisiae* was isolated a total of 81 times, which represents 56.6% of the total isolates: 28 isolates from the LD distillery (19.6%), 30 from ND (21%), and 23 isolates from NA samples (16%).

The second predominant species was *K. marxianus*, found in 30 isolates (21%) from the three distilleries: 13 isolates (9.1%) from LD samples, 5 (3.5%) from ND samples, and 12 (8.4%) from NA samples, which suggests a constant participation of the species in *A. durangensis* fermentation. From *Z. bailii*, 15 individuals were found, representing 10.5% of the isolates; *P. manshurica* corresponded to 5.6% of isolates (8 isolates); *P. kluyveri* and *T. delbrueckii* were found in 3 isolates each (2.1%); *H. guilliermondii* in 2 isolates (1.4%) and *C. lusitaniae* in 1 isolate, representing 0.7%. It is important to mention that *P. kluyveri* and *H. guilliermondii* were only detected in the NA distillery, and *C. lusitaniae* was found to be present only in ND. At the end of fermentation, in the spring season, *S. cerevisiae* and *P. manshurica* were found (Figure 3a), but only *S. cerevisiae* and *K. marxianus* were found in the LD distillery in the summer season (Figure 3b). In distillery ND, *T. delbrueckii* was detected at all sampling times during the spring season, but it was not present in the

summer season fermentation samples. On the other hand, fermentation in NA presented *S. cerevisiae*, *K. marxianus*, and *Z. bailii* in the spring and *S. cerevisiae*, *Z. bailii*, and *P. manshurica* in the summer season, which was less diverse in yeast species. All of this indicates that there are some yeast species that are dependent on the production season, probably due to temperature conditions, as observed for *T. delbrueckii* and *Pichia kluyveri*.

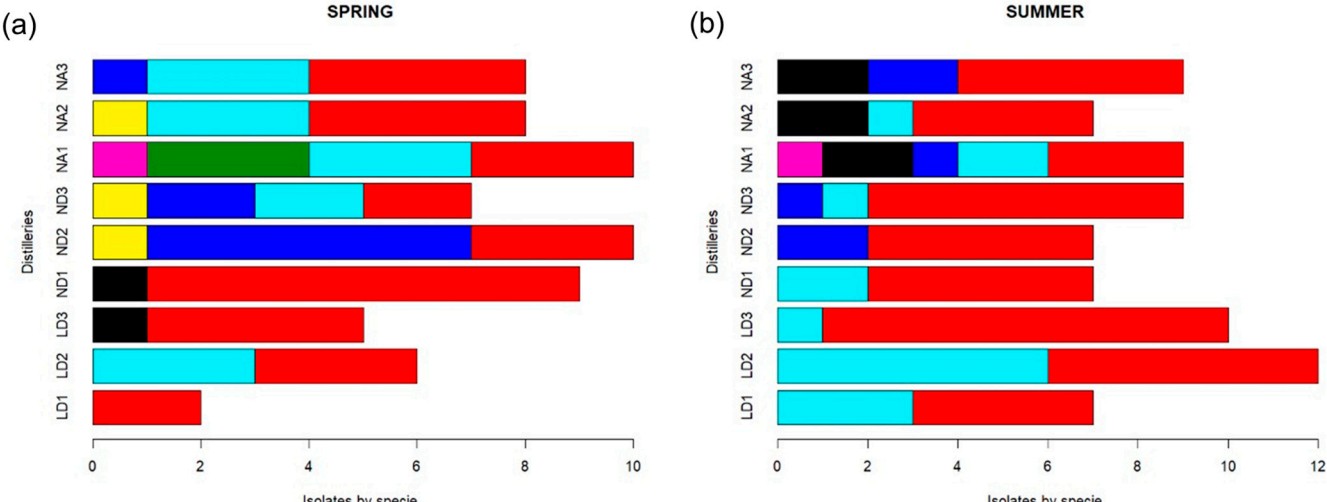

**Figure 3.** Frequency of yeast species isolated from mezcal fermentations from Durango State during the (**a**) spring season and (**b**) summer season. Distillery codes are as follows: Durango (LD), La Constancia (ND), and Nombre de Dios (NA). Numbers after distillery codes are as follows: beginning (1), middle (2), and end (3) of fermentation for each distillery: Yeast species are as follows: ■ *Saccharomyces cerevisiae*, ■ *Kluyveromyces marxianus*, ■ *Pichia manshurica*, ■ *Pichia kluyveri*, ■ *Torulaspora delbrueckii*, ■ *Hanseniaspora guilliermondii*, and ■ *Zygosaccharomyces bailii*.

### 3.2. Diversity Indices

According to the Simpson and Shannon–Wiener indices, the ND distillery presents the highest yeast diversity for both seasons, which coincides with the higher species richness denoted by the Margalef index (Figure 4). Remarkably, the diversity found in ND during the summer season is almost twice the diversity found in the spring season fermentation. In LD and NA, the Simpson index was similar for the two sampling seasons. LD presented the highest dominance indices (D) in both seasons.

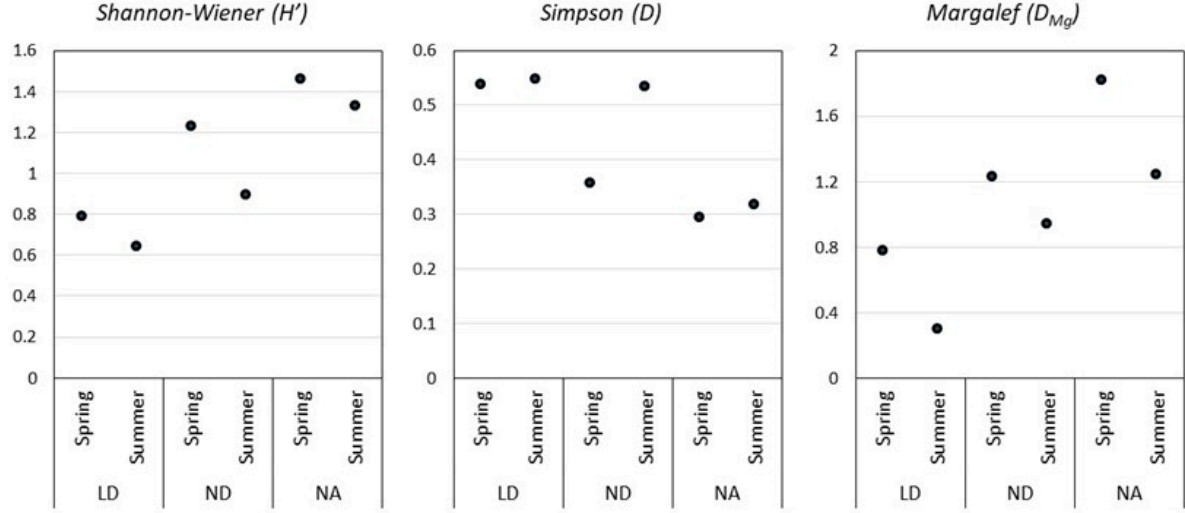

**Figure 4.** Diversity indices of the selected mezcal distilleries. Durango (LD), La Constancia (ND), and Nombre de Dios (NA).

Analysis of the diversity indices among the distilleries in the spring fermentation indicates significant differences in the Simpson indices from LD and NA. When analyzing the summer data, significant differences were found among LD and NA for the Simpson index and among LD against ND and NA for the Margalef index.

The PCA considering the isolates by genus showed differences in the distillery distribution according to the sampling season (Figure 5). PC1 and PC2 were selected to conduct this analysis, and the Cos2 values were considered to evaluate the quality and contribution of the sample variables. In the spring, the ND distillery was differentially grouped, particularly the ND2 (intermedia fermentation) and ND3 (final fermentation) samples. In the summer, the LD distillery was the most differentially grouped, particularly the LD2 (intermedia fermentation) and LD3 (final fermentation) samples, which suggested the microbial composition differences in these distilleries according to the season, mainly in the last stages of the fermentation process.

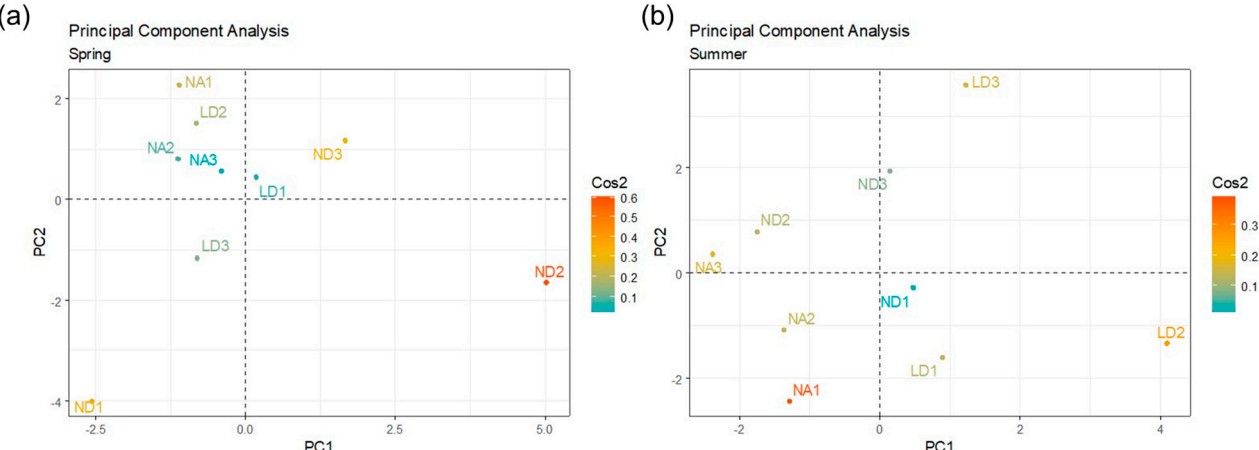

**Figure 5.** PCA considering the isolates by genus, and as variables, the different fermentation times in each distillery (NA, ND, and LD) within each sampling season: (**a**) spring or (**b**) summer. The quality and contribution of each variable was indicated by their Cos2 value (color gradient).

### 3.3. Fermentation Conditions

To determine the sampling season and production process effects on the must temperature, fermentation time, bacteria and yeast CFUs, as well as the must physicochemical conditions, an ANOVA was performed with data obtained from the response variables. Significant differences were tested with $p \leq 0.05$. Statistical analysis was performed through SAS 9.0.

The fermentation stage, density, pH, reducing sugars, refraction index, and solids measured during the fermentation process presented significant differences ($p \leq 0.05$) by the fermentation process (Table 2). The ANOVA performed to estimate the effect of the sampling temperature on the different response variables did not show significant differences.

**Table 2.** Effect of the production process in mezcal factories classified as semi-technified (LD, Durango) or artisanal (NA, Nombre de Dios) mezcal at the final sampling time.

| Process | Semi-Technified (LD) | Artisanal (NA) |
|---|---|---|
| Fermentation (d) | 6 | 2 |
| Must temperature (°C) | 32.83 | 30.75 |
| Bacteria LB (CFU) | $1.30 \times 10^7$ | $8.30 \times 10^6$ |
| Bacteria (CFU) | $2.30 \times 10^7$ | $1.70 \times 10^7$ |
| Yeast (CFU) | $1.10 \times 10^7$ | $9.40 \times 10^6$ |
| Protein (%) | 0.82 | 0.81 |

**Table 2.** *Cont.*

| Process | Semi-Technified (LD) | Artisanal (NA) |
|---|---|---|
| Density (g/cm³) | 1.03 | 1.01 |
| pH | 4.25 | 3.97 |
| Reducing sugars (mg/mL) | 10.46 | 4.25 |
| Refraction (°Brix) | 11.83 | 5.69 |
| Total solids (ppm) | 701.67 | 588.33 |

## 4. Discussion

Sequencing of the D1/D2 LSU region allowed the identification of yeasts at a species level. Eight species were found in the fermentation samples from the three mezcal-producing distilleries in Durango State, as follows: *S. cerevisiae*, *K. marxianus*, *Z. bailii*, *P. manshurica*, *P. kluyveri*, *T. delbrueckii*, *H. guilliermondii*, and *C. lusitaniae* (Figure 2). Yeast *S. cerevisiae* turned out to be the predominant one in both the spring and summer fermentations in the three producing distilleries, which coincides with what was previously reported in studies on the fermentation of *A. durangensis* in the State of Durango, as reported by [4], which analyzed two producing regions of the State of Durango, finding greater yeast diversity in the initial stages of fermentation; within the yeast species reported in these stages are *K. marxianus*, *T. delbrueckii*, and *C. diversa*. However, these authors indicated that, in the end, only *S. cerevisiae* was recovered from the region I, while region II exhibited a higher diversity of yeasts in the early stages of fermentation, as follows: *S. cerevisiae*, *K. marxianus*, *T. delbrueckii*, *C. diversa*, *P. fermentans*, and *H. uvarum*. Nevertheless, *T. delbrueckii* was found in a higher number than *S. cerevisiae* and *C. diversa* at the end of the fermentation. In the present study, there were similarities in some of the species, such as *P. manshurica*, *P. kluyveri*, *Z. bailii*, and *H. guilliermondii*, which were identified, but *C. diversa*, *P. fermentans*, and *H. uvarum* were not found.

Valdez et al. analyzed cooked *A. salmiana* fermentations in San Luis Potosi State and reported eight yeast species, as follows: *S. cerevisiae*, *K. marxianus*, *P. kluyveri*, *Z. bailii*, *C. lusitaniae*, *T. delbrueckii*, *C. ethanolica* and *S. exiguus* [11]. Six of them were identified in this research, plus *H. guilliermondii* and *P. manshurica*. Escalante-Minakata et al. found fewer yeast species in agave fermentation from San Luis Potosi State, as they reported only three species, *C. lusitaniae*, *K. marxianus*, and *P. fermentans* [10], probably related to the different agave species studied by these authors. On the other hand, Kirchmayr et al. analyzed two distilleries in Oaxaca State and found *S. cerevisiae*, *K. marxianus*, *Z. rouxii*, *Z. bisporus*, *T. delbreuckii*, and *Z. bisporus* to be the dominant yeasts; their results match those from this research, where dominant yeasts were *S. cerevisiae* and *K. marxianus* [7]. Kirchmayr et al. and Páez-Lerma et al. mentioned that species like *K. marxianus* and *T. delbrueckii* are associated with *S. cerevisiae* throughout the alcoholic fermentation [4,7]. It should be noted that *K. marxianus* has been reported as responsible for the production of aromatic compounds such as alcohols and esters, which give the beverage a fruity aroma [25]. Unlike the analyses carried out in mezcal fermentations, Lachance detected higher yeast diversity in tequila fermentation [26]. Yeasts reported in such research were *S. cerevisiae*, *Z. bailii*, *Candida milleri*, *Brettanomyces anomala*, *Brettanomyces bruxellensis*, *H. guilliermondii*, *Hanseniaspora vinae*, *P. membranaefaciens*, *T. delbrueckii*, and *K. marxianus*. Five of these yeasts were also found in our research.

Lachenmeier et al. mentioned that tequila is the beverage produced out of agave in Mexico with less composition variability; thus, variability found in mezcal could be considered a consequence of the agave species used for its production, as well as the local elaboration practices in the different regions with the denomination of origin and the differences in their technification levels in plant cultivation and processing [2]. In tequila production, only *Agave tequilana* is used, and highly industrialized production processes with few variables are concentrated in the authorized production zone [27]. The predominance of non-*Saccharomyces* yeasts could be explained by the temperature in the fermentation vats, which was around 40 °C. Non-*Saccharomyces* yeasts, such as

*K. marxianus*, are characterized by their ability to tolerate temperatures of approximately 40 °C [28]. Another parameter that could influence the species observed is the saponin content in *A. duranguensis*, which is high at 6.17 mg/g of dry tissue [29]; saponins work as yeasts growth inhibitors [29,30] and can affect the microbial composition of the agave must fermentation process. The presence of *P. manshurica* in the fermentation from Durango State is noteworthy, as this yeast has only been previously reported by Nolasco-Cancino et al. as a particular species of the State of Oaxaca [28].

Diversity indices (Figure 4) allowed us to establish that the Nombre de Dios distillery (NA), in the spring and in the summer season, was the one that presented the highest diversity of yeasts and the lowest dominance concentration, meaning there was no dominant species in this fermentation, which coincides with the Margalef index that denotes the higher species richness. On the contrary, the Durango distillery presented lower diversity in the spring season and higher *S. cerevisiae* dominance. However, the analysis of variance did not show significant differences in the diversity index (H') and the Margalef index (DMg) among the three distilleries in the spring season. The present work agrees with Enríquez-Salazar et al., who analyzed the microbial diversity in mead during different seasons, finding that, during spring and summer, there were higher indices of richness and biodiversity [13]. Kirchmayr et al. observed higher abundance indices (H' = 1.509–2.223) than the ones here reported (H' = 0.6442–1.332), given that they found higher species numbers during the fermentation process [7]. Several authors have reported that the qualitative and quantitative composition of the microbiota present throughout grape must fermentation depends mainly on factors such as the ethanol concentration, the region of origin of the raw material, the production process, the temperature, and the pH [31]. The ANOVA used here to determine the effect of the process over yeast and bacteria CFUs, as well as the must physicochemical characteristics, showed that the fermentation time, density, pH, reducing sugars, refraction index, and dissolved solids were significantly different ($p \leq 0.05$). The pH change explains this during fermentation because of the presence of organic acids and acetic acid, for example, generated through acetic fermentation performed by *Acetobacter* spp. bacteria [32]. Reducing sugars values suggest that fructooligosaccharides were not completely hydrolyzed during the cooking process, attributable to the use of rustic ovens in which temperature control is not possible [33]. Yeast *S. cerevisiae* has been reported to grow in environmental conditions present in fermentation vats, with a pH of 4 and high temperatures [4,7,11]. All of this is highly important, as the ethanol final concentration in the product is a function of the initial reducing sugars concentration and the loss of ethanol due to acetic acid generation [32].

In this study, eight different yeast species were found from samples taken at three fermentation stages (beginning, mid-fermentation, and end of fermentation) in three mezcal distilleries from Durango State, including *S. cerevisiae*, *K. marxianus*, *Z. bailii*, *T. delbrueckii*, *P. kluyveri*, *P. manshurica*, *H. guilliermondii*, and *C. lusitaniae*. By far, yeast *S. cerevisiae* was the most predominant in the three analyzed distilleries, since it was detected in all sampling sites, followed by *K. marxianus*. In Tamaulipas mezcal, *S. cerevisiae* strains also showed a high predominance, probably because of the must environmental conditions that increased the phenotype diversity of this species [34]. It is possible that mezcal fermentations in these two Mexican states (Durango and Tamaulipas) share a similar yeast diversity due to the traditional production practices and the use of some common *Agave* species, such as *A. americana* and *A. angustifolia*, as there are wild populations of these plant species along the northeast and north of Mexico.

The remaining yeast species were detected sporadically. Noteworthy, *P. manshurica* was found in mezcal distilleries from Durango State, even when it has been reported as being exclusively from Oaxaca State. Although this study did not analyze in detail the bacterial community, significant amounts of bacteria were detected during the process, which may be another important factor in the alcoholic fermentation of mezcal and its distinctive organoleptic characteristics. Regarding volatile metabolites production, it has been reported [33–35] that a high number of compounds are produced during agave must

fermentations, and some of them have been associated to the presence of *S. cerevisiae*, and some to non-*Saccharomyces* species [35], with isoamyl acetate, ethyl butyrate, 2-methyl-1-butanol, ethyl hexanoate, ethyl acetate, ethyl octanoate, ethyl dodecanoate, and phenyl ethyl acetate as the most commonly detected, besides acetaldehyde and the obvious predominance of ethanol. It is expected that fermented products from these three distilleries will have such volatile compounds, as indicated by the hedonic tests performed on such spirits. A comprehensive review of all the volatile compounds present in agave spirits has been presented by De la Torre-González et al. [36], as well as in other high-volume sales liquors produced from fruits, stalks, and grains, such as wine, tequila, and whisky, and all the common chemical families of volatile compounds shared by these fermented beverages are presented in Table 5.2 of that work. The core chemical compounds are in agreement with those reported by Lin et al. [37]; however, compounds formed during agave cooking, which is rich in oligofructans, lignin, and saponins, provide the must with Maillard compounds (caramel-like), vanilla, furfural, and higher alcohol notes that are not present in other fermentations, many of which are toxic to wine yeasts [35], hence the importance to continuously search for already adapted yeasts to perform agave must fermentations more efficiently.

## 5. Conclusions

The results here presented show that the technification level of a mezcal distillery has a clear influence, not only in the physicochemical parameter of the fermentation, but also in the culturable yeast diversity found. As the studied distilleries have no temperature control or isolation from external conditions, they can only operate in the mild seasons, which, in fact, also determines the type and abundance of the microorganisms isolated. The use of an inoculum in the Durango distillery affected the fermentation yeast diversity by reducing the abundance of species in both seasons, and this lower diversity also affected the final sugar consumption and pH of the fermented musts, having higher values than the semi-artisanal and artisanal distilleries. This could be related to the lack or insufficient concentration of non-*Saccharomyces* yeast that can consume the high fructose concentrations found in all agave musts. All of this suggests that improvements to the fermentation process can be still made regarding the productivity and quality of the final products of these mezcal producers, with the first step being the precise characterization of the yeast microbiota participating in their fermentation processes, aiming to manipulate their native inoculum, particularly for the artisanal distilleries.

**Author Contributions:** Conceptualization, S.C.M.-E., J.N.G.-R., C.P.L.-C. and I.C.-H.; Data curation, R.R.-H.; Formal analysis, I.C.-H.; Investigation, S.C.M.-E., J.A.N.-Z., R.R.-H., J.G.-Á. and J.N.G.-R.; Methodology, S.C.M.-E., J.A.N.-Z., J.G.-Á. and C.P.L.-C.; Resources, J.A.N.-Z., J.N.G.-R., C.P.L.-C. and I.C.-H.; Supervision, C.P.L.-C.; Validation, J.N.G.-R.; Writing—original draft, S.C.M.-E., J.A.N.-Z., R.R.-H. and I.C.-H.; Writing—review and editing, J.A.N.-Z., R.R.-H., J.N.G.-R. and C.P.L.-C. All authors have read and agreed to the published version of the manuscript.

**Funding:** This research was supported by the Instituto Politécnico Nacional (SIP-IPN), projects 2023-1440 and 2023-2851.

**Institutional Review Board Statement:** Not applicable.

**Informed Consent Statement:** Not applicable.

**Data Availability Statement:** All data are available in the manuscript.

**Conflicts of Interest:** The authors declare no conflicts of interest.

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
