# Peer review of "Diversity of Culturable Yeasts Associated with the Technification Level in the Process of Mezcal Production in the State of Durango"

_fermentation, doi:10.3390/fermentation10030147_

Round 1

Reviewer 1 Report

Comments and Suggestions for Authors

This article systematically studied the effects of brewing technology and seasonal changes on the microorganisms of mezcal, which has research value, but the following issues need to be carefully considered.

1. Line 127-128: The fermentation process is carried out in a wooden barrel container. When sampling 100 mL, did the different positions of the container are considered, such as the possible differences in the types and abundance of fermentation microorganisms in the upper, middle, and lower layers?

2. Line 196-227: The study should provide the results of direct extraction of total DNA from sampled samples for high-throughput sequencing, which can more intuitively reflect the relative abundance of microorganisms in fermentation samples. Identifying the species of colony through isolation and cultivation does not objectively characterize the composition of microorganisms during fermentation.

3. Representative strains isolated in this work should provide the storage number of an authoritative third-party storage institution.

4. This article did not conduct product flavor analysis and correlation analysis of microorganisms. The purpose of studying the differences and changes in microorganisms during the fermentation process is actually to illustrate their impact on the quality of fermented products, which should be carefully considered. Some recently published papers should be helpful for the analysis and discussion of this section, such as DOI: 10.1016/j.fbio.2023.103385.

Comments on the Quality of English Language

Moderate editing of English language required

Author Response

Comments and Suggestions for Authors

This article systematically studied the effects of brewing technology and seasonal changes on the microorganisms of mezcal, which has research value, but the following issues need to be carefully considered.

  1. Line 127-128: The fermentation process is carried out in a wooden barrel container. When sampling 100 mL, did the different positions of the container are considered, such as the possible differences in the types and abundance of fermentation microorganisms in the upper, middle, and lower layers?

Answer: An average sample of the upper, middle, and lower layers in the fermentation container was obtained, and the final sample is a combination of the three sampling positions. This was clarified on lines 131 and 132.

  1. Line 196-227: The study should provide the results of direct extraction of total DNA from sampled samples for high-throughput sequencing, which can more intuitively reflect the relative abundance of microorganisms in fermentation samples. Identifying the species of colony through isolation and cultivation does not objectively characterize the composition of microorganisms during fermentation.

Answer: We do agree with this comment, and indeed the total DNA extracted contained the whole microbial diversity, the project focused on yeasts to obtain new starter strains, that is why it was decided to isolate only cultivable yeasts, which are reported in this work. However, fungal (and bacterial) metagenomics is effectively missing, which is important in this type of analysis. The extraction of metagenomic DNA from agave musts presents some technical difficulties, due to its very astringent composition, and has been done by us, mainly for bacteria but from mezcal must obtained at other distilleries, not the ones analyzed in this work. Bacterial counts were obtained here only for quality control purposes, and they were not of interest for the producers as starters or any practical use. We indeed are working on obtaining such high-throughput efforts in a more controlled distillery.

  1. Representative strains isolated in this work should provide the storage number of an authoritative third-party storage institution.

Answer: We agree with the reviewer, therefore, we updated Figure 2 by adding the CBS code in the close relatives used in the analysis. Not all nearby sequences have CBS but we decided to keep them in the analysis because many have been previously isolated from other sources related to Agave musts, see table below. These nearby sequences were marked with an asterisk and their agave-related source in the corresponding figure caption. Finally, all study sequences are in GenBank (62 sequences, from MT322329.1 to MT322391.1 , hence the corresponding codes were updated in line 205 in the results section.

Closest relative, GenBank ID

CBS/Agave

KY108101

CBS:1560

JX141376

Agave Mezcal/Tamaulipas

KJ660860

N.A.

MT322857

Agave must/Tamaulipas

LC336455

N.A.

KT945088

N.A.

GU080052

N.A.

KY296086

Agave/Oaxaca

EU441913

N.A.

MH545926

CBS: 6936

MK211205

N.A.

JN391376

N.A.

HM191679

N.A.

MK034750

N.A.

MK394164

CBS:209

KY296079

Agave/Oaxaca

KF241560

N.A.

GQ249098

N.A.

KT945083

Agave must/Tamaulipas

KY296067

Agave/Oaxaca

MK394165

CBS:188

KY296072

Agave/Oaxaca

KT945091

Agave must/Tamaulipas

KP852496

N.A.

N.A. Not aplicable

Figure 2, Updated on line 214

  1. This article did not conduct product flavor analysis and correlation analysis of microorganisms. The purpose of studying the differences and changes in microorganisms during the fermentation process is actually to illustrate their impact on the quality of fermented products, which should be carefully considered. Some recently published papers should be helpful for the analysis and discussion of this section, such as DOI: 10.1016/j.fbio.2023.103385.

Answer:We reviewed the suggested article to be integrated in the discussion section (Lines 388 to 400) although the paper suggested do not deal with agave fermentations, hence, we have included some of our work (De la Torre-Gonzalez et al., 2017) were all information regarding microbial diversity and aromatic profiles of the main distilled beverages are presented, including agave distillates. Therefore the references were updated. We thank the reviewer for his/her valuable comments and suggestions.

Reviewer 2 Report

Comments and Suggestions for Authors

In this study, the authors typed the yeast microbiota during the production process of mezcal. Three distilleries which follow different production protocols were monitored. In general, the manuscript is well written, but some general points should be mentioned. The conclusion should be written carefully, as non Saccharomyces species may be more fructophilic but at the same time are more implicated in spoilage. This point of view should also be discussed. Additionally, it is not clear why they have isolated bacteria as they have not been typed or in general discussed. The statistical analysis is in-depth and well done, but the figures should be a little more explained. 

You can find here more comments: 

Line 22-23 

Better be precise by telling the exact month than the season 

Keyword 

You can add non Saccharomyces as well 

Introduction 

Write the hole name of the species when it is the first time that you mention it 

Line 89 

levels of technification? ..Maybe different production techniques?? I am not sure that your expression is correct 

Line 101 

Which commercial strain? Pls write it 

Figure 1 

Semi artisanal? You don’t use this term in the text before 

Line 131 

Pls precise the exact period of spring and summer  

2.2. Sample collection 

Pls be precise when the samples for yeast typing were chosen 

Line 144 

Why only 48h for bacteria? Add a reference 

Line 178 

Any ref for the Cos2 values? 

Line 197 

143 yeast colonies. Pls specify how many colonies/sample 

Figure 2 

Can you pls describe how you typed at strain level each species? 

Make a table in the Material and Methods with the reference strains 

Figure 3 

What is the 1,2,3? Explain more in the legend also the species. The color of the legend should not be the same between a and b 

Discussion 

It would be nice to discuss the fact that new species appear during fermentation for the 3 distilleries that were not present in the beginning, ex Td 

Why do you have isolated bacteria if you haven’t typed them? Pls comment more or erase their isolation 

Line 388 

Microbiota or yeast? 

Conclusion 

Can the indigenous microbiota have a negative impact on the production?

Author Response

Review Report Form #2

In this study, the authors typed the yeast microbiota during the production process of mezcal. Three distilleries which follow different production protocols were monitored. In general, the manuscript is well written, but some general points should be mentioned. The conclusion should be written carefully, as non Saccharomyces species may be more fructophilic but at the same time are more implicated in spoilage. This point of view should also be discussed. Additionally, it is not clear why they have isolated bacteria as they have not been typed or in general discussed. The statistical analysis is in-depth and well done, but the figures should be a little more explained.

Answer: We agree with the reviewer comment about the non-Saccharomyces species implicated in the spoilage of fermented beverages, however, as species-specific concentration of metabolic active populations was not quantified, it is hard to conclude their specific organoleptic implication on the aromatic profiles of the final products. We included more references about what is known in terms of volatile compounds for Agave spp fermentations (lines 388-400), which, apart from very technified tequila factories that make use of specific inocula and process control technology, all the rest are spontaneous, open-vessel fermentations, which widely vary on their microbial composition. The presence of bacteria in agave fermentations has been indeed reported, but it is usually used as a quality control (spoilage) parameter, due to acetic acid production at the end of the fermentation.

  • Line 22-23. Better be precise by telling the exact month than the season

Answer: We just specified a wide range, as each distillery do not operate at the same time, but depending on the schedule of the producers, hence for comparison purposes, we only indicate the season.

  • Keyword - You can add non Saccharomyces as well 

Answer: Done

  • Introduction  -Write the hole name of the species when it is the first time that you mention it Line 89 , levels of technification? Maybe different production techniques?? I am not sure that your expression is correct.

Answer: We have corrected to the whole name of a species when it appears for first time in the text, they are marked in yellow. The production technique is the same in the three distilleries, as we tried to show in Fig. 1, presenting the four main process steps (operational units), but what we mean by technification levels is how varies the methods and materials used to carry-out each process step. For example, for cooking, a rustic process is to use a pit in the ground with wood and hot stones, and a technified process is the use of an industrial, gas operated autoclave. As it can be seen in fig. 1, the distilleries have a mixture of rustic and technified steps, being the extremes, for a technified level, the use of stainless steel, electrical machinery and gas fuel, and for a rustic one, the use of wood for containers and for heating, manual or animal milling, and manual movement of liquids. This terms have been included in lines 98-99 and 101-102, besides the explanation given in lines 115-127.

  • Line 101 Which commercial strain? Pls write it 

Answer: We do not know the exact strain used by the producer, as it only stated that it was a commercial bread starter from a grocery store, without further information.

  • Figure 1 Semi artisanal? You don’t use this term in the text before

Answer: Agree, we have corrected this, and it is marked in yellow in line 99.

  • Line 131 Pls precise the exact period of spring and summer  

Answer: We just specified a wide range, as each distillery do not operate at the same time, but depending on the schedule of the producers, hence for comparison purposes, we only indicate the season, as understanding that spring goes from March to May and summer goes from June to August.

  • 2. Sample collection - Pls be precise when the samples for yeast typing were chosen

Answer: We included line 131-132 to precise what comprised a sample, and the times chosen for sampling depended when the producer allowed us to sample, with the indication that that was a start, middle or final time for their fermentations.

  • Line 144 Why only 48h for bacteria? Add a reference 

Answer: That is what the official norm states (NOM-092-SSA1-1994) this reference is included.

  • Line 178  Any ref for the Cos2 values? 

Answer: A high cos2 indicates a good representation of the variable on the principal component. In this case the variable is positioned close to the circumference of the correlation circle. A low cos2 indicates that the variable is not perfectly represented by the PCs. A better explanation of this is given in the section “Squared Cosine of a Component with an Observation” of follow reference, that was added

Abdi, H., & Williams, L. J. (2010). Principal component analysis. Wiley Interdisciplinary Reviews. Computational Statistics, 2(4), 433–459. https://doi.org/10.1002/wics.101

  • Line 197 143  yeast colonies. Pls specify how many colonies/sample

Answer: The number of colonies varied among samples, that is why we only reported the final selected ones, but certainly a very intense isolation/purification/selection process was carried out.

  • Figure 2- Can you pls describe how you typed at strain level each species? Make a table in the Material and Methods with the reference strains 

Answer: We updated figure 2 to include the CBS code in the reference strains and in cases where the strain was isolated from Agave we also added this in the name. We consequently update figure 2 and its figure caption. We are also registering all the sequences obtained in GenBank and this information will be added to the body of the MS text in the results part, line 205.

  • Figure 3 - What is the 1,2,3? Explain more in the legend also the species. The color of the legend should not be the same between a and b 

Answer: The legend, position of data and colors were corrected to make all the information indeed clearer.

  • Discussion - It would be nice to discuss the fact that new species appear during fermentation for the 3 distilleries that were not present in the beginning, ex Td 

Answer: T. delbrueckii and P. kluyveri are detected only on spring in two distilleries, we suppose that it is more sensitive to ambient temperature than the other yeast species, we included this observation in lines 232-234.

  • Why do you have isolated bacteria if you haven’t typed them? Pls comment more or erase their isolation

Answer: The presence of bacteria in agave fermentations has been indeed reported, but it is usually used as a quality control (spoilage) parameter, due to acetic acid production at the end of the fermentation. Bacteria were quantified as it is part of the official norm for this kind of products.

  • Line 388 - Microbiota or yeast? 

Answer: We have changed to yeast, it is highlighted in yellow, now line 408-409

  • Conclusion- Can the indigenous microbiota have a negative impact on the production?

Answer: It certainly could, but the demonstration for such negative impact would need to be assessed under more controlled conditions than the ones used in the three distilleries. Main interest of the producers is to stop the process as soon as the fermentation ends, and proceed with distillation, to avoid the increase of acetic acid on the fermented must. The isolated yeasts have been reported in other fermented beverages, and some of them have been associated to a better aromatic profile of the fermented product, as it is the case for K. marxianus and T. delbrueckii.

Round 2

Reviewer 1 Report

Comments and Suggestions for Authors

Most of the comments are properly responded. But one of the crucial question was not answered. The cultured colonies could not represent the real microbial composition during fermentation. The data are actually not valuable, such as Figure 3. A high throughput sequencing of the fermentation sample should be carefully considered. Correlation analysis between microbiota and flavor chemicals could be offered to give a more convincing conclusion.

Comments on the Quality of English Language

Minor editing of English language required

Author Response

Answer: We do agree with Reviewer #1 stating that the data presented is not the whole microbial composition, and indeed, a high-throughput sequencing analysis for both, bacterial and fungal DNA, would provide a complete picture of the microbial diversity of these fermentation systems, and we would indeed very much like to have the complete microbial metagenome of mezcal from Durango, but these samples, as previously stated, were focused on the aim of the project, that was to find new yeast starters, hence only cultivable yeasts were seek. Perhaps the title of our work should be corrected accordingly to clearly indicate this as:

Diversity of culturable yeasts associated with the technification level in the process of mezcal production in the state of Durango

We are currently working with mezcal musts from other States, and such analysis will be carried-out, but not for the samples presented in this work. As for the correlation of microbiota and flavor chemical composition, we can only speculate, based on what it is known for other Agave fermentations, as the specific volatile profiles were not assessed, hence we cannot make such correlation analysis, and preferred not to give a conclusion but just some clues.